# Facile Synthesis of Two-Dimensional Natural Vermiculite Films for High-Performance Solid-State Electrolytes

**DOI:** 10.3390/ma16020729

**Published:** 2023-01-11

**Authors:** Yan Xing, Xiaopeng Chen, Yujia Huang, Xiali Zhen, Lujun Wei, Xiqiang Zhong, Wei Pan

**Affiliations:** 1New Energy Technology Engineering Lab of Jiangsu Province, School of Science, Nanjing University of Posts & Telecommunications (NUPT), Nanjing 210023, China; 2State Key Lab of New Ceramics and Fine Processing, School of Materials Science and Engineering, Tsinghua University, Beijing 100084, China; 3School of Materials Science and Hydrogen Energy, Foshan University, Foshan 528000, China; 4Key Laboratory of Aerospace Advanced Materials and Performance of Ministry of Education, School of Materials Science and Engineering, Beihang University, Beijing 100191, China

**Keywords:** vermiculite, film, ionic exchanging, ionic conduction, ceramic electrolytes

## Abstract

Ceramic electrolytes hold application prospects in all-solid-state lithium batteries (ASSLB). However, the ionic conductivity of ceramic electrolytes is limited by their large thickness and intrinsic resistance. To cope with this challenge, a two-dimensional (2D) vermiculite film has been successfully prepared by self-assembling expanded vermiculite nanosheets. The raw vermiculite mineral is first exfoliated to thin sheets of several atomic layers with about 1.2 nm interlayer channels by a thermal expansion and ionic exchanging treatment. Then, through vacuum filtration, the ion-exchanged expanded vermiculite (IEVMT) sheets can be assembled into thin films with a controllable thickness. Benefiting from the thin thickness and naturally lamellar framework, the as-prepared IEVMT thin film exhibits excellent ionic conductivity of 0.310 S·cm^−1^ at 600 °C with low excitation energy. In addition, the IEVMT thin film demonstrates good mechanical and thermal stability with a low coefficient of friction of 0.51 and a low thermal conductivity of 3.9 × 10^−3^ W·m^−1^·K^−1^. This reveals that reducing the thickness and utilizing the framework is effective in increasing the ionic conductivity and provides a promising stable and low-cost candidate for high-performance solid electrolytes.

## 1. Introduction

All-solid-state lithium battery (ASSLB) is a promising candidate for next-generation batteries, serving various electric vehicles and electronic devices [1,2,3]. Such a bright future of the ASSLB benefits from the dramatically improved safety and energy density after the liquid electrolyte system is replaced by the solid-state electrolyte [4,5,6]. Therefore, the solid-state electrolyte is of vital importance for the ASSLB and directly affects its performance [7]. Polymer electrolytes and ceramic electrolytes are the main kinds of solid-state electrolytes that have been extensively studied. Polymer electrolytes are more suitable for flexible solid-state batteries, as it owns the advantages of high flexibility and easy production [8]. However, the ionic conductivity, mechanical strength, and stability of polymer electrolytes are unsatisfactory. On the other hand, ceramic electrolytes, which can provide high ionic conductivity, wide electrochemical windows, good thermal and chemical stability, high elastic moduli, and low electronic conductivity, have attracted continuous attention [9,10,11]. In recent years, Li_7_La_3_Zr_2_O_12_ (LLZO) [12], Li_1.4_Al_0.4_Ti_1.6_(PO_4_)_3_ (LATP) [13], Li_0.34_La_0.56_TiO_3_ (LLTO) [14], vermiculite-Li_0.33_La_0.557_TiO_3_/poly(ethylene oxide) (Vr-LLTO/PEO) [15], and Li_6_PS_5_Cl [16] are excellent candidates for solid-state electrolytes, and various strategies have been designed to improve their performance. However, desirable ionic conductivity that is comparable with liquid electrolytes of about 10^−2^ S·cm^−1^ has not yet been achieved in ceramic electrolytes.

The ionic conductivity of ceramic electrolytes is mainly restricted to their resistance and large thickness [5,17]. In recent years, grain resistance, interfacial resistance, and intrinsic resistance have been reduced by techniques, such as doping, framework, etc., and the performance of ceramic electrolytes is getting better [18]. Thinking of thickness, typically, the ceramic electrolyte pellets are prepared by hot pressing and cold pressing. The resulting pellets always exhibit large thickness (>200 μm) and low ionic conductivity of (10^−6^–10^−5^ S·cm^−1^) [19,20,21]. For instance, Wang et al. prepared reduced LLTO pellets by a hot-pressing method, which exhibit an ionic conductivity of 1.6 × 10^−5^ S·cm^−1^ [19]. In addition, the densities of ceramic electrolytes are usually several times higher than that of polymer electrolytes. For example, the density of PEO-lithium bis(trifluoromethanesulfonyl)imide (LiTFSI) is 0.93 g·cm^−3^, while the density of LLZO is 5.1 g·cm^−3^ [14]. Thus, the energy density of ASSLB using ceramic solid-state electrolytes also strongly relies on the thickness. To this end, various methods have been designed to reduce the thickness of ceramic electrolytes in order to obtain higher ionic conductivity and energy density. Through techniques, such as phase-inversion [22], atomic layer deposition [23], pulsed laser deposition [24], etc., the thickness of ceramic electrolytes can be reduced to a few microns. Jiang et al. prepared a 25 μm LLTO thin film by a tape-casting method and demonstrated an improved ionic conductivity of 2.0 × 10^−5^ S·cm^−1^ compared to LLTO pellets (200 μm, 9.6 × 10^−6^ S·cm^−1^) [14]. Although the thickness of ceramic electrolytes can be reduced and previous works are inspiring, the ionic conductivity is still low, as well as the high cost and complex production.

Two-dimensional (2D) materials have the potential to serve as ceramic electrolytes due to their low density, thin thickness, and easy processing [25,26,27]. Dong et al. used a lamellar MOF membrane to grow a LLTO electrolyte and obtained a 11 μm composite thin film which achieved a high ionic conductivity of 1.19 × 10^−4^ S·cm^−1^ [7]. However, making large-scale 2D films or composite membranes still faces the challenges of complex preparation technology, low yield, high cost, and ragged quality [28,29]. To cope with these obstacles and find potential ceramic electrolytes, 2D lamellar clays wake on stages. Among them, vermiculite (VMT) is the representative one because of its unique microstructure and has received continuous attention. For years various research has focused on the synthesis of materials and new composites using natural vermiculites. For instance, Giannelis et al. concerned the polymer-layered silicate nanocomposites, which have already been the pioneers in 2D nanostructured materials [30]. Avilés et al. synthesized nitrogen ceramics from a vermiculite-polyacrylonitrile intercalated compound by carbothermal reduction [31]. However, high-quality thin films made of pure vermiculites have rarely been investigated, especially for solid-state electrolytes application.

VMTs are naturally occurring minerals that are abundant in resource, low cost, and non-toxic [32,33]. Typically, the VMTs have a layered phyllosilicate structure that is crystallized in a monoclinic system (space group 2/*m*) [34]. The phyllosilicate structure is composed of alternate tetrahedral-octahedral-tetrahedral aluminosilicate layers [35]. Interestingly, the interlayer space of VMTs is quite different from other clays which contain two water layers and exchangeable cations, such as Na^+^, K^+^, Mg^2+^, and Ca^2+^ [36,37,38]. Such a special structure endows VMTs with many possibilities, especially for ceramic electrolytes. Through a simple rapid heating process, the VMTs can be expanded into nanosheets owning tens of atomic layers, which is suitable for assembling thin films. Lv et al. employed VMT as a framework and prepared large-size, oriented, and defect-free Vr-LLTO electrolytes. The resultant 15 μm thick Vr-LLTO electrolyte exhibited an ionic conductivity of 8.22 × 10^−4^ S·cm^−1^ and a compressive modulus of 1.24 GPa [17]. In addition, the VMT owns natural framework and interlayer channels. Numerous studies have proved that oriented structure and framework can facilitate the transport of Li^+^ ions and thus achieve a higher conductivity [18,39,40]. Moreover, the VMTs have a high interlayer ion capacity [41]. By ion exchange process, functional ions or groups can be inserted into the interlayer space conveniently. Although VMT has been used as a template to prepare composite ceramic electrolytes, the ionic conductivity of pure VMT has rarely been noticed and the synthesis of large-scale high-quality VMT film is still challenging. VTM nanosheets after thermal expansion can introduce Li+ ions into the interlayer channels followed by ionic exchange. Taking advantage of high ion concentration, thin thickness, and interlayer channel, if VMT can be fabricated into electrolyte films, it can be expected that the VMT films may have high ionic conductivity, good mechanical properties, and thermal stability.

In this work, a novel procedure is designed to exfoliate the vermiculite minerals and assemble them into functional films. Such a method is described and discussed in detail. Then, the morphology and microstructure of the as-prepared 2D ion-exchanged expanded vermiculite (IEVMTs) sheets and films are comprehensively characterized. The IEVMTs films demonstrate a thin thickness with a 2D framework and internal channels. In addition, the electrical property and stability of the IEVMTs films are studied and the mechanism for ion transport are discussed. Combined with the advantages of high ionic conductivity, good stability, and low cost, the IEVMTs films show broad application prospects in the energy and environment fields.

## 2. Materials and Methods

### 2.1. Materials

Natural vermiculites (Xinjiang, China) were expanded at 600 °C for 30 min at a heating rate of 10 °C·min^−1^, and then were milled to powders through 200 mesh sieves, which are labeled as EVMTs. Sodium chloride, hydrochloric acid, and lithium chloride were purchased from Sinopharm Chemical Reagent Co., Ltd., Beijing, China.

### 2.2. Synthesis of Ion-Exchanged Expanded Vermiculite (IEVMT) Sheets

The IEVMT sheets were prepared by a convenient two-step immersion method using the saturated solution [42,43]. The natural VMT powder (2g), after thermal expansion and sieving, was first dispersed in 50 mL of saturated NaCl solution. Then, the mixture was continuously stirred for 24 h under 50 °C to completely replace the interlayer ions with Na^+^. After the first step of exchanging, the Na-IEVMTs were separated by centrifugation from the mixture and washed several times with deionized water. After that, a second-step exchanging process was conducted by using LiCl solution under the same condition to displace Na^+^ with Li^+^. After filtering, the Li-IEVMT sheets were collected and then pickled in HCl solution (3 vol%) for 24 h to remove excess interstitial ions. Finally, the IEVMT powders were separated, washed with deionized water and ethanol, and dried overnight under a vacuum.

### 2.3. Preparation of IEVMT Films

The IEVMT films were prepared by a simple vacuum filtration method. The as-prepared IEVMT sheets (0.2 g) were dispersed in deionized water (100 mL) by ultrasonic treatment. Then, a light-yellow solution was obtained and dropwise added to a vacuum filtration system (−0.1 MPa). After the solvent was completely drained, a smooth and uniform thin IEVMT film was obtained and can be easily peeled off from the filter membrane. The IEVMT film was further dehydrated under room temperature overnight. Finally, an IEVMT film with a thickness of about 30 μm was successfully prepared.

### 2.4. Characterization

The phase constitution of the IEVMT product was analyzed by X-ray diffraction (XRD, Rigaku, Japan) by using Cu Kα radiation of 0.15418 nm. All the samples were scanned over the 2θ range from 5° to 50°. The morphology of IEVMT film was obtained on a field-emission scanning electron microscope (SEM, Merlin VP compact, Carl Zeiss, Oberkochen, Germany). TEM images were recorded by a JEOL-2010F transmission electron microscope (TEM) with an acceleration voltage of 200 kV. The thickness of the IEVMT sheet was measured by an atomic force microscope (AFM, Cypher, Oxford Instruments, MA, USA). The ionic conductivity of the film was investigated by impedance spectroscopy (IS, 1260A, Solartron, Leicestershire, UK) in the temperature range of 200–600 °C in air. The thermal diffusivity was measured by LFA-467 (NETZSCH, Selb, Germany). To measure the ionic conductivity, about 50 nm thick platinum film was sprayed on both sides of the film with a gap width of 2 mm to ensure good conductivity between silver electrodes and IEVMT film. Subsequently, a silver paste (HD100-75, Grikin, Beijing, China) was painted on both sides of the platinum surface with a silver wire buried in. After heating at 400 °C for 30 min, the silver electrode was consolidated and was available for IS tests. To conduct the IS tests, the IEVMT film with as-prepared silver electrodes was put into a muffle furnace, and the AC impedance spectrum was measured at different temperatures on an impedance analyzer coupled with a dielectric interface (IS, 1260A, Solartron, Leicestershire, UK) [44].

## 3. Results and Discussion

### 3.1. Synthesis and Characterization of IEVMT

Figure 1 illustrates the preparation process of the IEVMT films. As shown in Figure 1a, the raw VMT materials own natural lamellar structure which is composed of alternate tetrahedral-octahedral-tetrahedral aluminosilicate sheets. The interlayer space is filled with H_2_O molecules and cations. The strong van der Waals forces lead to a narrow interlayer distance of raw VMT minerals. After heat treatment, the rapid evaporation of H_2_O dramatically debilitates the combination force between layers, looses the tight lamellar structure, and expands the interlayer space as shown in Figure 1b. The expansion of interlayer distance facilitates the next ion-exchanging step. The Na^+^ and Li^+^ cations are successively exchanged into the interlayer to replace the original ions, such as K^+^, Ca+, and Mg^+^ (Figure 1c,d) [45]. The ion exchanging treatment further increases the layer spacing, which contributes to faster ions transport. After heat treatment and ionic exchange process, the raw VMT minerals can be exfoliated to thin nanosheets with a larger interlayer distance. For preparing the thin IEVMT films, a simple filtration process is employed as shown in Figure 1e. With the aid of pressure difference, the 2D IEVMT sheets are separated from the solution, dried, and self-assembled into thin films. As shown in Figure 1f, the as-prepared IEVMT film is self-supporting and can be directly removed from the filter membrane. Finally, a uniform and smooth thin film composed of natural vermiculite mineral is successfully synthesized by thermal expansion and ionic exchanging treatment.

The morphology of vermiculite in different steps was characterized to further confirm the synthetic process. Figure 2a shows the optical photo of the raw vermiculite minerals. The pristine vermiculite exhibits a scale-like morphology. When the vermiculite is rapidly heated at a heating rate of 10 °C·min^−1^, the evaporation of internal water forces the silicate layers apart, forming an elongated concertina-like particle (Figure 2b). The thickness of EVMT is twenty to thirty times larger than the raw materials. After a two-step ionic exchanging process, the interlayer cations have been displaced by Li^+^, leading to the increase in layer spacing and the exfoliation of the layers. The IEVMT sheets can disperse homogenously in water and form a light-yellow solution (Figure 2c), indicating that the IEVMT sheets are light, thin, and uniform. Figure 2d shows the SEM images of IEVMT sheets in that solution, demonstrating a lateral dimension of several microns. Through vacuum filtration, a paper-like thin film can be facilely assembled by using the dispersion shown in Figure 2c. The size and thickness of the IEVMT films are determined by the filtration device and the concentration of the solution. For example, Figure 2e,f show a thin IEVMT film with a diameter of 5 cm and a lamellar microstructure.

Figure 3 shows the TEM images of vermiculite sheets before and after ionic exchange. As shown in Figure 3a, the EVMT that undergoes thermal treatment still exhibits a book-like or concertina-like microstructure with a relatively large thickness. The EVMT sheets are stacked by several thin layers. After the ion exchanging process, the thickness of vermiculite sheets decreases remarkably to an almost single layer. The IEVMT sheet shown in Figure 3b is a typical 2D microstructure with a large area and smooth surface. The thickness of the IEVMT sheet is further characterized by an atomic force microscopy (AFM). The measured thickness of IEVMT is 3.75 nm and 1.55 nm, indicating a bilayer and single layer, respectively [46]. By the way, smooth AFM curves indicate that the roughness of the IEVMT film is relatively low. The TEM and AFM results confirm that the ionic exchanging process can further enlarge the interlayer space and exfoliate the raw vermiculite mineral into a 2D thin sheet with only a single or several atomic layers.

To further determine the crystal structure of the vermiculite products, X-ray diffraction (XRD) studies were conducted, as shown in Figure 4. The main peaks at 2θ of 8.6°, 27.02°, and 45.6° can be ascribed to (001), (003), and (005) diffraction planes of vermiculite, respectively, matching well with the standard card (JCPDS #42-1399). Thus, the {001} *d*-spacing of expanded vermiculite powder is 1.01 nm [47]. After the ionic exchanging process, the shift of {001} diffraction peak to low angle direction is observed, demonstrating a larger interlayer distance. The diffraction peak of IEVMT at 7.4° represents an interlayer spacing of 1.20 nm, which is consistent with the AFM results and indicates successful exfoliation. Therefore, all the characterization results confirm that vermiculite mineral can be reconstructed by the thermal expansion and ionic exchanging procedure, the obtained IEVMT sheet owns a single-layer overlapped microstructure instead of a multilayer stack. Moreover, the thin film assembled by IEVMT sheets is smooth, flexible, tailorable, and can be mass-produced, which shows a wide application prospect in ceramic electrolytes.

### 3.2. Electrical Properties of the IEVMT Film

The electrical transport properties of IEVMT film have a significant influence on its application. To explore the ionic conductivity of IEVMT film, an impedance spectroscope (IS) was employed. For IS characterization, the film needs to be made as an electrode. As shown in Figure 5a, a silver paste was used to form a silver electrode and fix the film at the quartz substrate. Before that, a Pt film was sprayed on the IEVMT film first to enhance the contact between the silver electrode and IEVMT film, while using a mask to reserve a 2 mm gap. When the electrode was ready, the sample was transferred into a muffle furnace, and the AC impedance spectrum was measured at different temperatures. As shown in Figure 5b, the Nyquist plots at different temperatures all show a semicircle shape, and the radius equals the resistance. As the temperature increases, the semicircle becomes smaller, indicating a decrease in the resistance of the IEVMT film. A model of *R*-CPE unit (resistor-capacitor in parallel) and R connected in series is used as the equivalent circuit to fit the spectra, as shown in the inset of Figure 5b. Figure 5c shows the conductivity of the IEVMT films measured at different temperatures. As can be seen, the ionic conductivity of the IEVMT film increases from 0.023 to 0.310 S·cm^−1^ in the temperature range from 450 to 600 °C. The ionic conductivity of the IEVMT thin film demonstrates a great improvement on previous works, such as AO/10ScSZ films [40], Vr-LLTO/PEO [15], Vr-LLTO [17], etc. According to the Arrhenius equation, the activation energy (*E*_a_) of Li^+^ ions in the IEVMT film is about 0.855 eV, which is very low for silicate and oxide materials, indicating that the migration of ions requires lower energy (Figure 5d) [40,48]. The decreased ionic conductivity and activation energy indicate a faster transport process of Li^+^ ions inside the IEVMT films, which is mainly attributed to the larger interlayer channel and rigid framework. Due to thermal expansion and ionic exchanging process, the interlayer space of IEVMT is more enlarged than that of raw vermiculite materials. The transport channel for lithium ions becomes wider and the drag force is decreased. In addition, combined with the rigid framework of vermiculite nanosheet, the intrinsic resistance is dramatically reduced [7,17]. Therefore, the ionic conductivity of IEVMT films in the horizontal direction shows a great improvement. In particular, for the ceramic electrolyte application, the ionic conductivity in the medium temperature range of 400–700 °C is of great importance. The IEVMT film with higher ionic conductivity shows wide application prospects in solid electrolytes at medium and high temperatures.

For superior ceramic electrolytes, a lower dielectric constant and dielectric loss is also essential. The dielectric constant and loss of IEVMT film were then tested and presented in Figure 6. As shown, the dielectric constant (*ε*_r_) of IEVMT film shows a peak at around 90 °C which refers to the ferroelectric-to-paraelectric phase transition temperature. The maximum dielectric constant is about 6.31 around 90 °C at 1 kHz and the minimum value is 2.25 at 10 MHz. The dielectric loss at a low frequency usually depends on the grain boundary polarization, and then dipolar rotation takes effect in the high-frequency range [49]. Thus, the increase in *ε*_r_ at lower frequencies is mainly due to the block of electrode-electrolyte polarization at the interface, while the rotation of carriers lags behind the change in electric field, resulting in the reduction in dielectric constant at a high-frequency range. Moreover, the enhanced Li^+^ ion mobility through the wide gap between IEVMT sheets under thermal activation may also lower the dielectric constant [50,51]. Surprisingly, the dielectric loss (tan *δ*, Figure 6b) of the IEVMT film maintains a low level (10^−2^–10^−1^) over a large range of temperatures, demonstrating that the increase in interlayer spacing reduces the friction between dipole molecules and therefore the polarization loss.

### 3.3. The Stability of IEVMT Film

The stability of ceramic electrolytes, meanwhile, will also be a prime concern. Usually, organic materials and polymers, such as rubber, have excellent wear resistance properties. Beyond that, 2D materials, such as graphene and MoS_2_, have excellent wear resistance properties and demonstrate application prospects in electrical contact materials [52]. Owing to the lamellar microstructure and stable mineral composition, the IEVMT film is expected to have good anti-wear performance. The coefficient of friction of the IEVMT film was characterized and compared with other 2D materials. As shown in Figure 7, the average coefficient of friction for IEVMT film is only 0.51, which is comparable with organic polyester [53] and much lower than that of other inorganic 2D materials, such as TiN [54], TiB_2_ [54], Mica [55], montmorillonite [56], MoS_2_/Cu-Zn composite [52], and graphene-based GNSs/SBR composite [57]. Moreover, the measured thermal diffusivity (α) of IEVMT film is 0.22 mm^2^·s^−1^ at 500 °C. According to the calculation formula for thermal conductivity *k* = *α* × *C*_p_ × *ρ*, where *k*, *α*, *C*_p_, and *ρ* represent the thermal conductivity, thermal diffusivity, heat capacity, and density, respectively, the *k* of IEVMT film is 3.9 × 10^−3^ W·m^−1^·K^−1^. The low thermal conductivity further facilities the application of IEVMT films in middle- and high-temperature environments. Therefore, the good anti-wear and thermal resistance properties of IEVMT films provide desirable stability and durability, ensuring the safety of long-term use.

## 4. Conclusions

In this work, a smooth, flexible, and free-standing thin IEVMT film is successfully prepared by using natural vermiculite minerals as starting materials, which provides a promising solution for the limitation of thickness and intrinsic resistance of ceramic electrolytes. A simple procedure, including thermal expansion, ion exchange, and vacuum filtration, is designed to exfoliate the raw vermiculite. The raw vermiculite mineral is first expanded by rapid heat treatment, in which the fast evaporation of internal water reduces the strong combination force between layers and expands the interlayer space. Followed by a two-step ionic exchange process, the vermiculites further exfoliate into thin nanosheets with one or two atomic layers. Then, through vacuum filtration, the IEVMT thin film is self-assembled by the above-exfoliated vermiculite sheets to 30 μm thin ceramic films, dramatically reducing the thickness of ceramic electrolytes. The detailed structure and morphology characterization confirms that single-layer vermiculite frameworks are obtained with a 1.2 nm interlayer channel, which can provide the transport channel for lithium ions and decrease the intrinsic resistance. High ionic conductivity of 0.31 S·cm^−1^ is achieved by the resulting IEVMT thin films. Simultaneously, a lower dielectric constant and dielectric loss are observed in the IEVMT films. Moreover, the IEVMT films exhibit a coefficient of friction of 0.51, which is lower than various 2D materials, and a low thermal conductivity of 3.9 × 10^−3^ W·m^−1^·K^−1^, demonstrating excellent stability. Combining the structure and properties analyses, the rigid nanosheets of vermiculites ensure stability, while the thin thickness and interlayer channel contribute to improved ionic conductivity. This superior comprehensive performance endows the IEVMT films with wide application prospects in medium- and high-temperature solid electrolytes.

## Figures and Tables

**Figure 1 materials-16-00729-f001:**
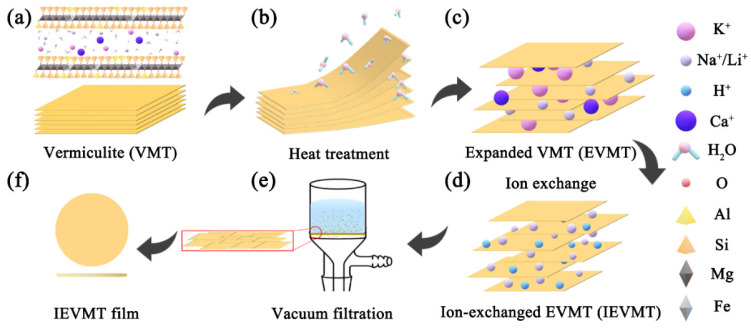
The schematic diagram of the preparation of IEVMT film. (**a**) the structure of vermiculite; (**b**) the heat treatment process; (**c**) the structural diagram of EVMT; (**d**) the ion exchanging treatment; (**e**) vacuum filtration; and (**f**) the IEVMT film.

**Figure 2 materials-16-00729-f002:**
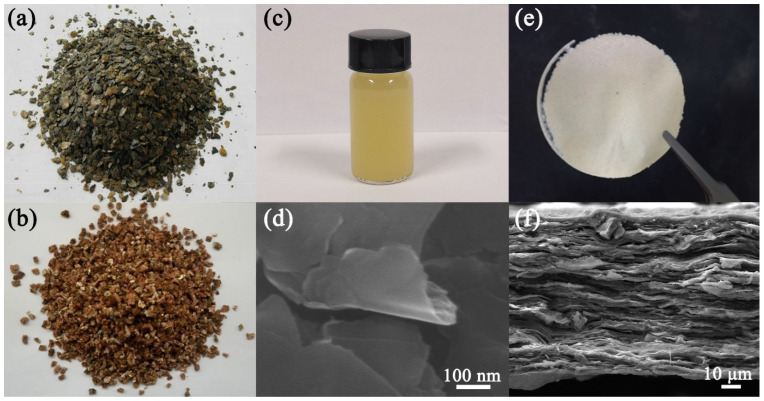
The morphology of vermiculite and film. The optical photos of raw vermiculite (**a**), thermal expanded vermiculite (**b**), IEVMT dispersion (**c**), and IEVMT film (**e**); the SEM images of dispersed IEVMT sheets (**d**) and the cross-section of IEVMT film (**f**).

**Figure 3 materials-16-00729-f003:**
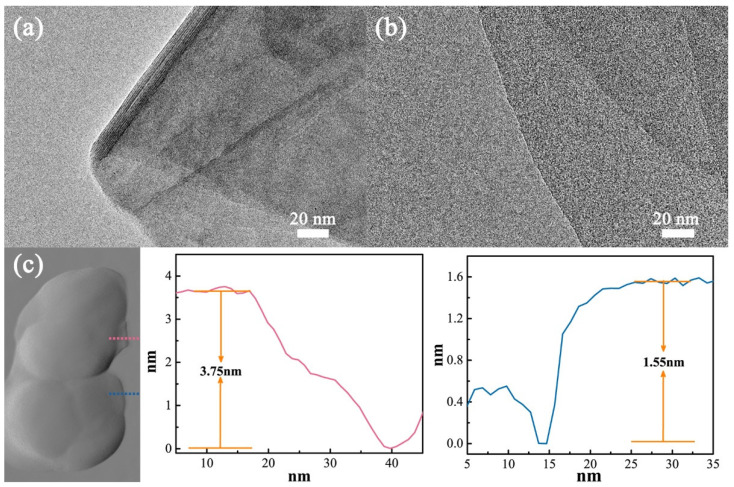
TEM images of the EVMT sheets (**a**) and IEVMT sheets (**b**), and the AFM characterization of IEVMT (**c**).

**Figure 4 materials-16-00729-f004:**
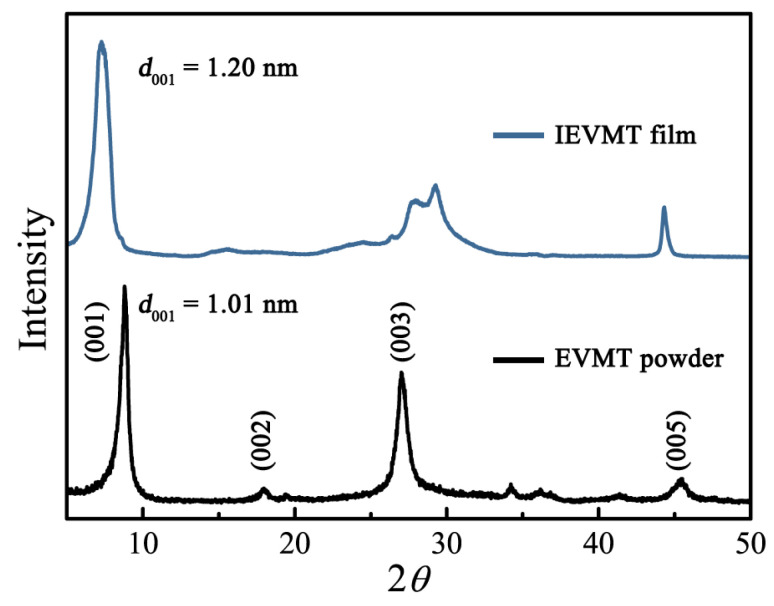
XRD patterns of the EVMT powder and the films after ion exchange.

**Figure 5 materials-16-00729-f005:**
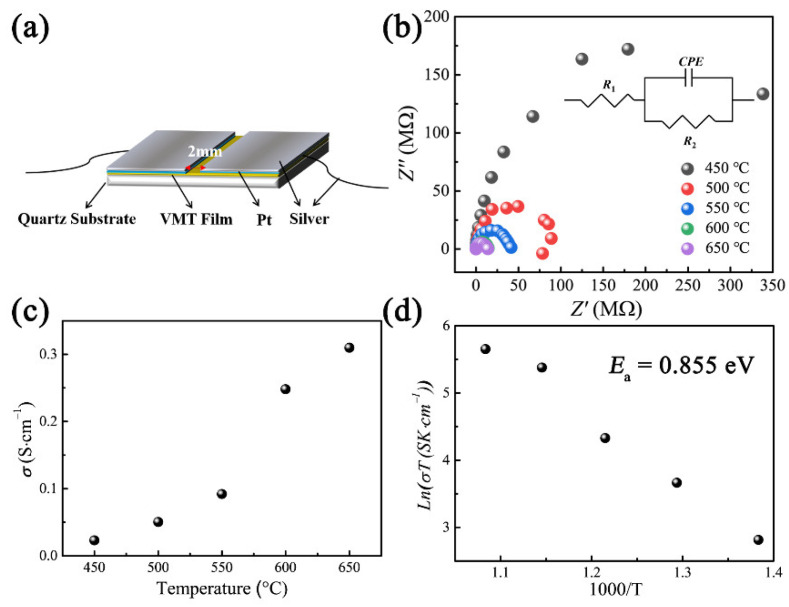
Schematic illustration of the device for the electrical measurement of the IEVMT film (**a**), Nyquist plots (**b**), ionic conductivity (**c**), and Arrhenius plot (**d**) of IEVMT film.

**Figure 6 materials-16-00729-f006:**
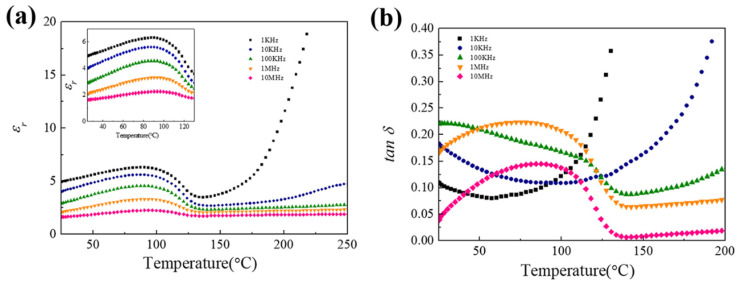
The dielectric constant (**a**) and dielectric loss (**b**) of IEVMT film at different temperatures in the frequency range of 1 kHz to10 MHz.

**Figure 7 materials-16-00729-f007:**
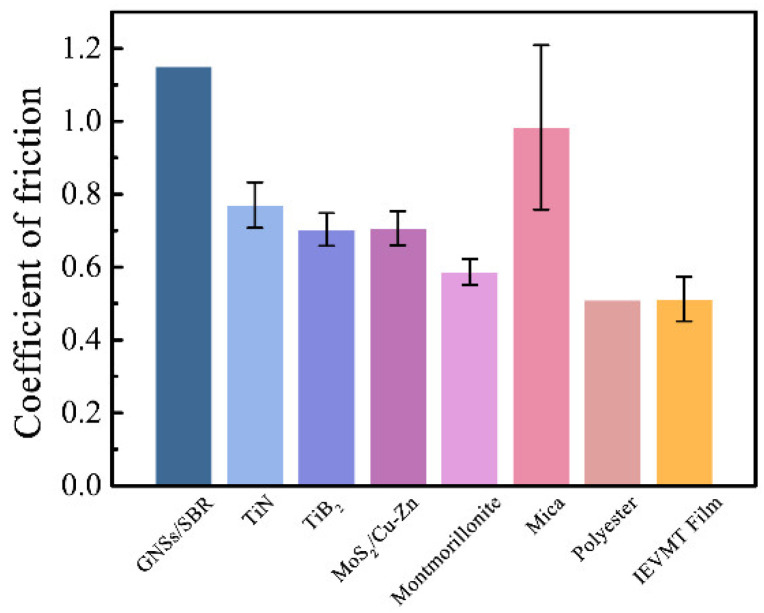
The anti-wear property of IEVMT film compared with other 2D materials.

## Data Availability

The data presented in this study are available on request from the corresponding author.

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
