# Peer review of "Facile Synthesis of Two-Dimensional Natural Vermiculite Films for High-Performance Solid-State Electrolytes"

_materials, 2023, doi:10.3390/ma16020729_

Round 1
Reviewer 1 Report
Comments: after a careful observation the manuscript need a Major change after which it can be accepted.
1. Authors have to rewrite whole introduction part, as most of the sentences are unfinished which needs to be refurbished. Eg. Line 29 2D materials are hotspot from past 30 years because of their physicochemical properties . . . ! what is your point in this ? why are you choosing ? point of necessity ? authors must focus on these terms.
2. Authors must use single form of notations through out the manuscript many mistakes are like these Eg: line 39 (T) -octa or (O)- tetra either of only one notation must be used.
3. Figure 2. Two times (b) was reported check
4. More explanation must be given why the sheet like structure was obtained with scientific briefing.
5. Through AFM authors must validate the roughness of the prepared film and to prove better contact angle analysis must also be added for the current experiment.
6. In Fig 4 authors must mention hkl values for the peaks obtained in the figure for easy understanding with JCPDS card number as reference or by plotting with reference.
7. In figure 5 b authors must add the circuit diagram for the EIS measurement. And explain in brief using Nernst equation. And the image caption mist be changed as it is not suiting the images. Also, authors are requested to add the graphs instead of the plot.
8. Error bars must be added for Fig.7.
9. It is requested to add a brief explanation with respect to elasticity, durability, and stability of the film.
10. Authors must add recent studies and compare the outcome of the perspective with much deeper elaborations. Also, authors can site some recent publications,
Hwa, K. Y., Ganguly, A., Santhan, A., & Sharma, T. S. K. (2022). Synthesis of Water-Soluble Cadmium Selenide/Zinc Sulfide Quantum Dots on Functionalized Multiwalled Carbon Nanotubes for Efficient Covalent Synergism in Determining Environmental Hazardous Phenolic Compounds. ACS Sustainable Chemistry & Engineering, 10(3), 1298-1315.
Jana, J., Van Phuc, T., Chung, J. S., Choi, W. M., & Hur, S. H. (2022). Nano-Dimensional Carbon Nanosphere Supported Non-Precious Metal Oxide Composite: A Cathode Material for Sea Water Reduction. Nanomaterials, 12(23), 4348.
Author Response
Dear Academic Editor and Reviewers:
Thank you for your attention and the comments on our manuscript entitled “Facile Synthesis of Two-dimensional Natural Vermiculite Films for High-Performance Solid-State Electrolytes” (Manuscript ID: materials-2117464) submitted to Materials.
We have substantially revised the manuscript according to the reviewer’s and editor’s comments. Please find our point-to-point responses to reviewers’ comments.
We declare that there are no conflicts of interest.
According to the guidance, we prepared and uploaded a marked copy of the revised manuscript that shows changes made on revision clearly highlighted for your convenience.
Many thanks again for all your time and efforts in processing/editing our manuscript, we hope you will find the current version much improved and meet the standard to be published in Materials.
We are looking forward to hearing from you soon!
Yours sincerely
Dr. Yan Xing
New Energy Technology Engineering Lab of Jiangsu Province
School of Science
Nanjing University of Posts & Telecommunications (NUPT)
E-mail: xingyan618@njupt.edu.cn
Responses to Reviewer 1
Point 1: Authors have to rewrite whole introduction part, as most of the sentences are unfinished which needs to be refurbished. Eg. Line 29 2D materials are hotspot from past 30 years because of their physicochemical properties . . . ! what is your point in this ? why are you choosing ? point of necessity ? authors must focus on these terms.
Response 1: Thank you for your constructive suggestion! We have substantially rewritten the introduction section. English grammar has been checked by a native English writer throughout the manuscript. In addition, the motivation of our work, the specific gap in the ceramic electrolyte, and the point of necessity have all been supplemented and discussed in the revised manuscript.
Point 2: Authors must use single form of notations throughout the manuscript many mistakes are like these Eg: line 39 (T) -octa or (O)- tetra either of only one notation must be used.
Response 2: Thank you for the kind suggestion. We have checked and corrected these mistakes throughout the revised manuscript.
Point 3: Figure 2. Two times (b) was reported check
Response 3: Thank you for the kind suggestion. We have corrected the labels in Figure 2.
Point 4: More explanation must be given why the sheet like structure was obtained with scientific briefing.
Response 4: Thank you for the kind suggestion. We have added the explanation about the formation of vermiculite sheets in section 3.1 in the revised manuscript. The raw VMT materials own natural lamellar structure which is composed of alternate tetrahedral-octahedral-tetrahedral aluminosilicate sheets. The interlayer space is filled with H2O molecules and cations. The strong van der Waals forces lead to a narrow interlayer distance of raw VMT minerals. After heat treatment, the rapid evaporation of H2O dramatically debilitates the combination force between layers, looses the tight lamellar structure, and expands the interlayer space as shown in Figure 1b. The expansion of interlayer distance facilitates the next ion-exchanging step. The Na+ and Li+ cations are successively exchanged into the interlayer to replace the original ions, such as K+, Ca+, and Mg+ (Figures 1c and 1d). The ion exchanging treatment further increases the layer spacing, which contributes to faster ions transport. After heat treatment and ionic exchange process, the raw VMT minerals can be exfoliated to thin nanosheets with a larger interlayer distance. The above explanation has been supplemented in the revised manuscript.
Point 5: Through AFM authors must validate the roughness of the prepared film and to prove better contact angle analysis must also be added for the current experiment.
Response 5: Thank you for your suggestion. The smooth AFM curves indicate that the roughness of the IEVMT film is relatively low. In addition, due to the water-absorbing characteristics and large interlayer space of vermiculates, the vermiculate sheets and films can be entirely wetted by water, thus the contact angle is 0°. We have provided the discussion about the roughness and wetting properties of the IEVMT films in the revised manuscript.
Point 6: In Fig 4 authors must mention hkl values for the peaks obtained in the figure for easy understanding with JCPDS card number as reference or by plotting with reference.
Response 6: Thank you for the kind suggestion. The main peaks at 2θ of 8.6°, 27.02°, and 45.6° can be ascribed to (001), (003), and (005) diffraction planes of vermiculite, matching well with the standard card (JCPDS #42-1399). We have supplemented the hkl values and standard JCPDS card in the revised manuscript.
Point 7: In figure 5 b authors must add the circuit diagram for the EIS measurement. And explain in brief using Nernst equation. And the image caption mist be changed as it is not suiting the images. Also, authors are requested to add the graphs instead of the plot.
Response 7: Thank you for the kind suggestion. A model of R-CPE unit (resistor-capacitor in parallel) and R connected in series is used as the equivalent circuit to fit the spectra and has been added inset of Figure 5b. Nernst equation does not apply to the results of the Impedance spectroscope, so it has not been discussed. Moreover, the caption of Figure 5 has been corrected.
Point 8: Error bars must be added for Fig.7.
Response 8: Thank you for your suggestion. Error bars have been added in Figure 7. To explain, in Figure 7, the coefficient of friction of the IEVMT film was characterized and compared with other 2D materials. The data of 2D materials such as TiN, TiB2, Mica, montmorillonite, MoS2/Cu-Zn composite, and graphene-based GNSs/SBR composite are all obtained from related published works, and not all references contain error bars. We have tried our best to supplement the available error bar data in Figure 7.
Point 9: It is requested to add a brief explanation with respect to elasticity, durability, and stability of the film.
Response 9: Thank you for your suggestion. The explanation of the elasticity, durability, and stability of the IEVMT films has been added in the revised manuscript.
Point 10: Authors must add recent studies and compare the outcome of the perspective with much deeper elaborations. Also, authors can site some recent publications,
Hwa, K. Y., Ganguly, A., Santhan, A., & Sharma, T. S. K. (2022). Synthesis of Water-Soluble Cadmium Selenide/Zinc Sulfide Quantum Dots on Functionalized Multiwalled Carbon Nanotubes for Efficient Covalent Synergism in Determining Environmental Hazardous Phenolic Compounds. ACS Sustainable Chemistry & Engineering, 10(3), 1298-1315.
Jana, J., Van Phuc, T., Chung, J. S., Choi, W. M., & Hur, S. H. (2022). Nano-Dimensional Carbon Nanosphere Supported Non-Precious Metal Oxide Composite: A Cathode Material for Sea Water Reduction. Nanomaterials, 12(23), 4348.
Response 10: Thank you for your constructive suggestion! We have supplemented recent studies and cited these publications.

Reviewer 2 Report
I have read and evaluated the manuscript and in my opinion, the submission does not yet sufficiently justify publication. In order to fix this problem, the addition of a description of recent development in the field of the research topic with citing recent comprehensive papers would be important in the introduction part. Discuss the shortcomings of previous work and the gaps and how this work intends to fill those gaps.
More and proper discussion about the results it is necessary.
The conclusion is also not targeted to the important aspects described in the manuscript; please rephrase it.
From the abstract and conclusions, I can only see the importance of the Natural two-dimensional (2D) materials, the results. No innovative ideas, experimental methods, and conclusions that can guide theoretical improvement or practical applications can be found.
The results must be compared with more studies.
Authors should also proofread their manuscript (some spelling and grammar errors). English grammar errors should be eliminated throughout the manuscript. A thorough checking from a native English writes would be a good idea.
The motivation part of the study is not exactly as clear as mentioned. Can you summarise the study by stating the primary research question?
Why should this study be detailed? In the continuation, the answers to the questions I mentioned below should be written clearly.- Do you consider the topic original or relevant in the field? Does it address a specific gap in the field?
- What does it add to the subject area compared with other published material?
- The general expression mentioned in both the abstract and the conclusion sections is used. More scientific and numerical results should be given.
Author Response
Dear Academic Editor and Reviewers:
Thank you for your attention and the comments on our manuscript entitled “Facile Synthesis of Two-dimensional Natural Vermiculite Films for High-Performance Solid-State Electrolytes” (Manuscript ID: materials-2117464) submitted to Materials.
We have substantially revised the manuscript according to the reviewer’s and editor’s comments. Please find our point-to-point responses to reviewers’ comments.
We declare that there are no conflicts of interest.
According to the guidance, we prepared and uploaded a marked copy of the revised manuscript that shows changes made on revision clearly highlighted for your convenience.
Many thanks again for all your time and efforts in processing/editing our manuscript, we hope you will find the current version much improved and meet the standard to be published in Materials.
We are looking forward to hearing from you soon!
Yours sincerely
Dr. Yan Xing
New Energy Technology Engineering Lab of Jiangsu Province
School of Science
Nanjing University of Posts & Telecommunications (NUPT)
E-mail: xingyan618@njupt.edu.cn
Responses to Reviewer 2
Point 1: I have read and evaluated the manuscript and in my opinion, the submission does not yet sufficiently justify publication. In order to fix this problem, the addition of a description of recent development in the field of the research topic with citing recent comprehensive papers would be important in the introduction part. Discuss the shortcomings of previous work and the gaps and how this work intends to fill those gaps.
Response 1: Thank you for your constructive suggestion! We have rewritten the Introduction section and substantially revised the manuscript.
Point 2: More and proper discussion about the results it is necessary.
Response 2: Thank you for your suggestion! We have supplemented adequate discussion about the synthesis process, microstructure, electrical properties, and stability in the revised manuscript.
Point 3: The conclusion is also not targeted to the important aspects described in the manuscript; please rephrase it.
Response 3: Thank you for the kind suggestion. We have rewritten the conclusion section and have paid more attention to describing the important idea and results in the revised manuscript.
Point 4: From the abstract and conclusions, I can only see the importance of the Natural two-dimensional (2D) materials, the results. No innovative ideas, experimental methods, and conclusions that can guide theoretical improvement or practical applications can be found. The results must be compared with more studies.
Response 4: Thank you for your constructive suggestion! We have rewritten the abstract and conclusion section according to your wise guidance. The innovative ideas, special preparation technology, and desirable conclusion have been described in detail and compared with other studies. Moreover, the theoretical improvement of ionic conductivity contributes to the thin film thickness and lamellar framework. The practical applications of prepared films have also been discussed based on the electrical properties, mechanical and thermal stabilities.
Point 5: Authors should also proofread their manuscript (some spelling and grammar errors). English grammar errors should be eliminated throughout the manuscript. A thorough checking from a native English writes would be a good idea.
Response 5: Thank you for the kind suggestion! We have sought help from a native English speaker and checked the English grammar throughout the manuscript.
Point 6: The motivation part of the study is not exactly as clear as mentioned. Can you summarise the study by stating the primary research question? Why should this study be detailed? In the continuation, the answers to the questions I mentioned below should be written clearly.
Do you consider the topic original or relevant in the field? Does it address a specific gap in the field?
What does it add to the subject area compared with other published material?
The general expression mentioned in both the abstract and the conclusion sections is used. More scientific and numerical results should be given.
Response 6: Thank you so much for the kind suggestion! According to your guidance, we have reorganized and rewritten the introduction section. We have started the introduction by concerning the primary research question in all-solid-state lithium batteries (ASSLB) and solid-state electrolytes. Addressing the limitation in current ceramic electrolytes and comparing it with other published work, we propose the novelty in our work and demonstrate that the novel vermiculite films (IEVMT) may provide a solution to plug the gap in the field of ceramic electrolytes. In addition, scientific and numerical results have been added in the abstract and conclusion sections. Of course, this is a brief outline. Please refer to the revised manuscript for details.

Reviewer 3 Report
In this work, the authors report their success in coping with the main problems in developing the ionic conductivity of ceramic electrolytes: thickness and intrinsic resistance. They prepared via self-assembling expanded vermiculite nanosheets. This manuscript requires minor revisions before being considered for publication in Materials or any journal in this field
1. The studied variables are not explicitly mentioned
2. [lines 185-186] the statement is unclear because it does not compare with the slow heating rate or what heating rate is used in this study?
3. Generally, this manuscript tends only to report the results of their investigation. There are few comparisons with the results of previous similar studies. Please mention the successes and improvements from the results of previous studies
4. The conclusion has not explicitly answered the purpose of the research to tackle the limitation of ceramic electrolytes as ionic conductivity
Author Response
Dear Academic Editor and Reviewers:
Thank you for your attention and the comments on our manuscript entitled “Facile Synthesis of Two-dimensional Natural Vermiculite Films for High-Performance Solid-State Electrolytes” (Manuscript ID: materials-2117464) submitted to Materials.
We have substantially revised the manuscript according to the reviewer’s and editor’s comments. Please find our point-to-point responses to reviewers’ comments.
We declare that there are no conflicts of interest.
According to the guidance, we prepared and uploaded a marked copy of the revised manuscript that shows changes made on revision clearly highlighted for your convenience.
Many thanks again for all your time and efforts in processing/editing our manuscript, we hope you will find the current version much improved and meet the standard to be published in Materials.
We are looking forward to hearing from you soon!
Yours sincerely
Dr. Yan Xing
New Energy Technology Engineering Lab of Jiangsu Province
School of Science
Nanjing University of Posts & Telecommunications (NUPT)
E-mail: xingyan618@njupt.edu.cn
Responses to Reviewer
In this work, the authors report their success in coping with the main problems in developing the ionic conductivity of ceramic electrolytes: thickness and intrinsic resistance. They prepared via self-assembling expanded vermiculite nanosheets. This manuscript requires minor revisions before being considered for publication in Materials or any journal in this field
Point 1: The studied variables are not explicitly mentioned
Response 1: Thank you for the kind suggestion. For the synthesis process, the kind of ions in the interlayer space is the variable. After a two-step ionic exchanging process, the original interlayer ions, such as K+, Ca+, and Mg+, have been replaced by Li+. When evaluating the ionic conductivity, the temperature is the variable. The ionic conductivity has been measured under different working temperatures, as shown in Figure 5. Thanks again for your suggestion, and we will try more variables in the subsequent research.
Point 2: [lines 185-186] the statement is unclear because it does not compare with the slow heating rate or what heating rate is used in this study?
Response 2: Thank you for your constructive suggestion! We have supplemented the heating rate (10 ℃·min−1) in line 186 and the experimental section 2.1.
Point 3: Generally, this manuscript tends only to report the results of their investigation. There are few comparisons with the results of previous similar studies. Please mention the successes and improvements from the results of previous studies
Response 3: Thank you for the kind suggestion. We have supplemented the comparison with other works. In lines 248-250, the ionic conductivity of IEVMT films has been compared with similar studies such as AO/10ScSZ films [40], Vr-LLTO/PEO [15], Vr-LLTO [17], etc. In lines 292-296, the friction performance of IEVMT films has been compared with other inorganic 2D materials like TiN [54], TiB2 [54], Mica [55], montmorillonite [56], MoS2/Cu-Zn composite [52], and graphene-based GNSs/SBR composite [57].
Point 4: The conclusion has not explicitly answered the purpose of the research to tackle the limitation of ceramic electrolytes as ionic conductivity
Response 4: Thank you for the kind suggestion. We have revised the conclusion section and added the purposed of our research.
Round 2
Reviewer 1 Report
manuscript is accepted in the current form
Author Response
Many thanks again for all your time and efforts in reviewing our manuscript!
Reviewer 2 Report
The study has been considerably improved compared to its previous format and it has been clearly stated that it has been made publishable by emphasizing the scientific outputs.
In this context, it is appropriate to publish the study as it is.
Author Response

(The authors gave the same response as above.)
